# Hepatoprotective and Antioxidant Effects of Nanopiperine against Cypermethrin via Mitigation of Oxidative Stress, Inflammations and Gene Expression Using qRT-PCR

**DOI:** 10.3390/ijms242015361

**Published:** 2023-10-19

**Authors:** Sohail Hussain, Abdulmajeed M. Jali, Saeed Alshahrani, Khairat H. M. Khairat, Rahimullah Siddiqui, Mohammad Intakhab Alam, Raisuddin Ali, Manal Mohammed, Andleeb Khan, Hamad Al Shahi, Ali Hanbashi, Marwa Qadri, Mohammad Ashafaq

**Affiliations:** 1Department of Pharmacology and Toxicology, College of Pharmacy, Jazan University, Jazan 45142, Saudi Arabia; shussainamu@gmail.com (S.H.); amjali@jazanu.edu.sa (A.M.J.); saalshahrani@jazanu.edu.sa (S.A.); khk1413@hotmail.com (K.H.M.K.); rahimullahsiddiqui@gmail.com (R.S.); halshahi@jazanu.edu.sa (H.A.S.); ahanbashi@jazanu.edu.sa (A.H.); mqadri@jazanu.edu.sa (M.Q.); 2Maternity and Children Hospital, Najran 66243, Saudi Arabia; 3Department of Pharmaceutics, College of Pharmacy, Jazan University, Jazan 45142, Saudi Arabia; intak4u@yahoo.co.in; 4Department of Pharmaceutics, College of Pharmacy, King Saud University, Riyadh 11451, Saudi Arabia; ramohammad@ksu.edu.sa; 5Substance Abuse Research Center (SARC), College of Pharmacy, Jazan University, Jazan 45142, Saudi Arabia; manlroa@yahoo.com; 6Department of Biosciences, Faculty of Science, Integral University, Lucknow 226026, India; drandleebkhan@gmail.com; 7Department of Pharmacology, University of Oxford, Mansfield Road, Oxford OX1 3QT, UK; 8Inflammation Pharmacology and Drug Discovery Unit, Medical Research Center (MRC), Jazan University, Jazan 45142, Saudi Arabia

**Keywords:** antioxidants, reactive oxygen species, nanoparticles, cytokines, RT-PCR, cypermethrin, oxidative stress, cytokines, hepatotoxicity

## Abstract

Cypermethrin (Cyp) is a pyrethroid that has been associated with the toxicity of various organs. The aim of our study was to evaluate the hepatoprotective and antioxidant activities of nano-piperine (NP) against Cyp toxicity. Cyp (50 mg/kg) was administered orally in all animals of groups III–VI for 15 days. Groups IV–VI each received three doses of NP (125, 250, and 500 µg/kg/day) for 10 days after receiving the Cyp dosage, which was given after 1 h. A rise in serum biomarkers (ALT, AST, ALP, total protein, and albumin), which are indicators of toxicity alongside anomalous oxidative stress indices (lipid peroxidation (LPO), glutathione (GSH), superoxide dismutase (SOD) and catalase), was detected. After Cyp treatment, we observed upregulated cytokines, caspase expression, and histological analysis that the showed distortion of cell shape. However, the administration of NP dramatically reversed all of the Cyp-induced alterations, inducing reductions in serum marker levels, stress level, the production of cytokines, and caspase expression. Additionally, all of the histopathological alterations were minimized to values that were comparable to normal levels. The present findings suggested that NP exhibits potent antioxidant and anti-inflammatory activities that can protect rats’ livers against Cyp-induced liver damage through hepatoprotective activities.

## 1. Introduction

Liver disease is a significant health concern in the United States, with over 100 million Americans diagnosed. Among these, nonalcoholic fatty liver disease (NAFLD), commonly referred to as fatty liver, affects between 80 and 100 million individuals. In the US alone, liver disease has been identified in 4.5 million people, accounting for 1.8% of the population. If left untreated, liver disease can escalate to liver cancer and liver failure. In 2020, liver disease was responsible for 51,642 adult deaths in the US, translating to a rate of 15.7 per 100,000 [1,2].

Pesticides application is expected to rise and inflict potential effects on the health of nontarget species, including human health. Human malignancies have been related to pesticide use [2,3]. Pesticides are widely used in the agricultural and horticultural industries, and most people are exposed to them through their diet [4,5]. Pesticide degradation is a very slow process [6,7], which may make it easier for people to be exposed through skin contact and ingestion [8]. Rodents develop liver tumors and hepatocellular carcinoma as a result of exposure to organochlorines, especially dichlorodiphenyltrichloroethane (DDT) and dichlorodiphenyl dichloroethylene 54 (DDE) [9,10].

The most widely used insecticide is Cyp, one of numerous environmental pollutants. It has been used for both residential and commercial purposes because of its potent insecticidal effects [11,12]. It has great biodegradability, is less toxic, and is especially effective against a variety of insects. There have also been some instances of animal toxicity [13,14]. Cyp is toxic and has previously been reported to possess links to hepato-renal dysfunction, testicular cancer, motor activity, and neurotoxic effects due to its ability to pass the blood–brain barrier [15,16,17].

Liver enzymes are primarily responsible for metabolizing Cyp, which is regarded by the liver as a secondary target organ. Cyp metabolites may induce oxidative stress by damaging hepatic cells and releasing reactive oxygen species (ROS) [18]. Additionally, oxidative stress causes DNA damage, protein oxidation, and oxidative damage to lipid content [19], which further activates liver inflammation and apoptosis [20]. The activation of liver inflammation and apoptosis by oxidative stress results in further DNA damage, protein oxidation, and oxidative damage to lipid content [19,20].

Researchers are now concentrating on the beneficial effects of nutraceuticals. Another benefit of employing nutraceuticals to treat free radical toxicity is that they are less costly and come with fewer adverse effects than current medicine [21]. Black pepper’s piperine, which has pharmacological qualities such as causing antioxidant, bioenhancer, anti-inflammatory, and hepatoprotective actions [22,23], was selected for use in the current study. Our earlier research established the cardioprotective properties of nanopiperine [24]. The antioxidative effects of piperine are due to its 5-membered ring structure [25].

The mechanism of action of piperine inhibits the enzymes that break down drugs, promotes absorption by activating gut amino acid transporters, blocks the cell pump that removes drugs from cells, and blocks intestine glucuronic acid synthesis. When a medicine goes through the liver after being absorbed from the GIT, it may inhibit enzymes involved in drug metabolism or boost GIT absorption of the drug. It also interferes with the hepatic enzymes UDP glucuronyltransferase and aryl hydrocarbon hydroxylase, which are involved in the drug metabolism [26]. Cytochrome P450 3A4 (CYP3A4) and human P-glycoprotein are both inhibited by piperine [27]. Both proteins play significant roles in the first-pass removal of many medications.

Recent technical developments have encouraged researchers to use nanoencapsulation devices. Due to their small size, which makes it simple for cell components to absorb them, these systems have considerable potential for employment as carriers of bioactive substances [28]. As a result, nanoformulation opens the door to a future with lower mortality. We developed piperine nanoparticles for the current study because there is a dearth of information on the hepatoprotective properties of nanopiperine against Cyp toxicity.

To determine the hepatotoxicity of cypermethrin, we evaluated biochemical parameters, antioxidant status, inflammatory cytokines, qualitative and quantitative DNA determination, and histological abnormalities associated with liver damage.

## 2. Results

### 2.1. Particle Size, Polydispersity Index, and Zeta Potential

Using lipids and surfactants, hot homogenization methods were used for NP formulation. As illustrated in (Figure 1), the TEM analysis of NP revealed that its form was spherical. Zetasizer was used to examine the values for the polydispersity index (PDI), zeta potential, and particle size distribution. It was revealed that NP had an average size of 136.6 ± 20.4 nm (*n* = 3), showing the PDI to be 0.297 ± 0.062 (*n* = 3) (Figure 2). The surface charge on the NP was 27.9 ± 9.18 mV (*n* = 3) (negative value). 

### 2.2. Blood Serum Assay

Blood serum markers (ALT, AST, and ALP) significantly rose (*p* < 0.01; *p* < 0.001) under Cyp (group III) treatment compared to the control (group I). However, we successfully cut down the raised levels (*p* < 0.05, *p* < 0.01, *p* < 0.001) of these markers after co-treatment with NP (group IV–V). There were no changes observed in the group treated with NP alone (Group II) in contrast to the control. Total protein and albumin decreased in the Cyp group, while co-treatment with NP ameliorated all the parameters (Table 1).

### 2.3. Effects on LPO and GSH

Malondialdehyde (MDA) levels significantly increased (*p* < 0.001) after exposure to Cyp, while GSH levels significantly (*p* < 0.001) decreased (Table 2). In contrast, group IV–V rats receiving Cyp supplemented with NP showed significant effects (*p* < 0.05; *p* < 0.01; *p* < 0.001) and LPO and GSH levels were ameliorated.

### 2.4. Effects on SOD and Catalase

There was a marked decrease in the activity of SOD and catalase (*p* < 0.001; *p* < 0.01) in the liver of group III (Table 3) compared to the control. Conversely, NP treatment resulted in the amelioration of SOD and created a catalase near to the control (*p* < 0.05; *p* < 0.01). 

### 2.5. Caspases Assay

Cyp alone (group III) had an increase (*p* < 0.001) in expression of caspase-3 and 9 caused by Cyp. Additionally, in the groups IV and V co-treatment with NP significantly (*p* < 0.05, *p* < 0.01, and *p* < 0.001) suppressed the expressions (Figure 3 and Figure 4).

### 2.6. Inflammatory Markers

The use of Cyp alone (group III) amplified (*p* < 0.001) the levels of cytokines (IL-1β, IL-6, and TNF-α) (Table 3). Furthermore, the co-treatment of NP reversed all the changes in the expression levels of these cytokines from groups IV to V (*p* < 0.05, *p* < 0.01, *p* < 0.001). The expression in group II was nearly the equal to that in the control.

### 2.7. DNA Fragmentation Produced by Cyp

Cyp treatment prompted DNA damage, which is critical evidence of program cell death. DNA fragmentation assay results obtained in the control and treatment groups (Cyp and NP) demonstrated genomic DNA damage patterns (Figure 5). Like the control, the Cyp group had visible DNA fragmentation. In the Cyp + NP (250 and 500 µg/kg/day) group, less DNA smearing was seen, indicating DNA protection. In the NP-only group, no DNA smearing was found. The percentage of the anti-apoptosis effect of NP in protecting DNA integrity against Cyp exposure was also estimated. An amplified apoptotic percentage was seen in Cyp (267%, *p* < 0.001). As compared to the Cyp, NP co-administration (250 and 500 µg) effectively reduced DNA fragmentation to 207% and 150% (*p* < 0.01; *p* < 0.001 respectively) (Figure 6). 

### 2.8. Histopathological Studies

A typical healthy architecture of liver cells was observed in the control (Figure 7) and the NP-treated group (slide not shown) showed the same cell architecture as the control. The control slide represented normal hepatic cells, clear sinusoidal spaces, and a central vein. However, various cellular changes were noticed in the Cyp-treated group, such as a severe parenchymal architecturally disrupted parenchymal layer, congestion in the central vein, and enlarged sinusoids. These architectural alterations were protected when seen after NP treatment in the Cyp group (Cyp + NP 250 and 500) in contrast to the Cyp-only group in a dose-dependent manner. The Cyp + NP group exhibited fever changes and a clear central vein close to the control group. Figure 8 shows our results according to severity score.

### 2.9. Effect of NP on RNA Expression

qRT-PCR was used to confirm the inflammatory and apoptotic expression of mRNA (Figure 9). According to our results, Cyp treatment considerably increased the expression of Bax, NF-kB, IL-1, IL-6, and caspases 3 and 9 when linked to the control group (*p* < 0.001). NP supplementation noticeably lowered the level of inflammatory and apoptotic responses dose-dependently (*p* < 0.05, *p* < 0.01 and *p* < 0.001). After being compared to the control, only NP failed to demonstrate any effects. 

## 3. Discussion

Our study provides robust evidence supporting the hepatoprotective effects of nano-piperine against Cyp-induced liver toxicity in rats. Cellular antioxidative defense mechanisms are vulnerable to reactive oxygen species (ROS) produced by Cyp, leading to potential damage to cellular macromolecules. Our experimental findings corroborate that Cyp treatment instigates an oxidative imbalance within the liver. Given these circumstances, piperine emerges as a promising therapeutic agent to counter liver toxicity. Previous studies have indicated that the toxicity mediated by Cyp results in an elevated concentration of ROS within cells, culminating in oxidative stress [29].

The oxidative damage instigated by ROS manifests in the form of elevated levels of AST, ALT, ALP, and a concomitant decrease in total protein and albumin activity. These markers, indicative of tissue damage, are subsequently released into the bloodstream. Predominantly located in the liver, ALT, when released into the blood, serves as a critical marker. Our observations of increased serum ALT and AST levels align with the findings of a previous study by Yousef et al. [30]. Cyp exposure was found to precipitate a decline in total protein and albumin levels, a phenomenon also reported by Yousef et al. They posited that the reduction in total protein predominantly stemmed from a decrease in albumin rather than the globulin fraction [31]. Rivarola and Balengo concluded that the use of pesticides may be the cause of the drop in plasma protein, particularly in albumin. According to Rivarola and Balengo’s assessment, changes in the liver’s protein metabolism may be the cause for the decrease in plasma protein caused by pesticide treatment, particularly in albumin [32]. Consequently, the oscillating serum albumin levels serve as pivotal indicators, shedding light on the severity, progression, and prognosis of liver disorders.

Lipid peroxidation (LPO) is instrumental in the onset of various diseases. It involves membrane permeability, leading to enzyme leakage [32]. At an early stage of LPO, free radicals also trigger cellular damage by deactivating membrane enzymes, depolymerizing polysaccharides, and breaking down proteins. Prior research has documented that exposure to pyrethroids inflicts tissue damage in both rats and rabbits [33]. 

Our findings unequivocally demonstrate that liver LPO levels increased significantly as a result of cypermethrin poisoning, while antioxidant enzyme (SOD, CAT) activities decreased. Cyp exposure led to decreased GSH levels in the liver. Depleted GSH resulted in spurts in the generation of ROS and ended with an oxidant and antioxidant imbalance that resulted in cell death [34]. Furthermore, tissues had lower levels of GSH. Treatment with NP at various doses significantly decreased LPO as measured by MDA, and the co-treated groups also had an increase in GSH levels. Piperine’s lipophilic nature may have slowed down the process of lipid peroxidation that produces free radical production, reducing cell damage [35]. Furthermore, the propensity of piperine to enhance membrane permeability may have facilitated the absorption of GSH into cells, amplifying the activity of the GSH transport system and bolstering the intracellular antioxidant status [36].

Antioxidant enzymes, such as SOD and catalase, are considered to constittue the primary defense against oxidative damage. They play a pivotal role in thwarting the formation of free radicals [35]. The surge in ROS production, leading to an escalation in LPO, results in the attenuation of SOD and catalase enzymes. Following cutaneous exposure to Cyp, we observed a discernible decline in the activities of both SOD and catalase enzymes within tissues [37], accompanied by a reduction in GSH [38]. SOD, a metallocenzyme, facilitates the catalytic conversion of superoxide anions into molecular oxygen and water. Catalase, a haem-enzyme, neutralizes the detrimental effects of hydrogen peroxide by converting it into non-toxic oxygen and water. In animals treated with Cyp, a decline in catalase was observed, implicating its role in the breakdown of hydrogen peroxide. It has also been postulated that LPO might significantly contribute to the observed reduction in catalase activity in the context of cypermethrin toxicity [38]. NP administration successfully attenuated the catalase and SOD activity when compared to the results for Cyp-only groups.

The intricate mechanisms through which ROS activates both inflammatory and apoptotic indices remain a subject of ongoing research. Our findings elucidate that ROS production stimulates the release of TNF-α, IL-1β, and IL-6, in tandem with caspases 3 and 9. These polypeptides exert multifaceted effects on cells and modulate the gene expression essential for protective mechanisms and immune responses [39,40]. Our current findings underscore that Cyp-induced excessive production of TNF-alpha, IL-1, IL-6, caspases 3, and 9 precipitates hepatotoxicity. This aligns with prior research that showcased the potential of piperine to mitigate Cyp-induced dysfunction across various organs [24,41].

DNA fragmentation patterns serve as a reliable metric with which to evaluate apoptosis [29]. In cell death, endonucleases and proteolytic enzymes fragmented DNA and the chromatin into a smear-like pattern [42]. In the control group (lane-1), the absence of a DNA laddering pattern, indicative of no DNA fragmentation, was observed. The DNA remained confined close to the well, presenting as a compact band—a pattern consistent with the results from the NP-only group. In contrast, the Cyp-treated group (third lane) exhibited DNA laddering in the form of a smear, indicative of significant DNA fragmentation. The smear appearance diminished in a dose-dependent manner in the co-treated groups. At higher NP doses (500 μg/kg/day), the reduced smearing pattern (fifth lane) was akin to that shown by the control. Quantitative DNA damage, in relation to percentage apoptosis, further corroborated our findings.

Histopathological evaluations revealed a healthy liver cell architecture in both the control (Figure 7) and NP-treated groups. The control slide depicted normal cells with distinct sinusoidal spaces and a central vein. In stark contrast, the Cyp-treated group exhibited multiple cellular aberrations, including a disrupted parenchymal layer, central vein congestion, and dilated sinusoids. These architectural perturbations were effectively mitigated following NP treatment in the Cyp group, aligning with prior findings by Sabina et al. [43].

## 4. Materials and Method

### 4.1. Chemicals

Interleukin assay kits for IL-1β, IL-6, TNF-α, and caspase-3, and 9 were obtained from Abcam (Cambridge, UK) and cypermethrin and piperine were received from Sigma Chemicals located in Balcatta, WA, (USA). Primers were bought from Macrogen Inc., Seoul, Republic of Korea, while RNA isolation and cDNA and qRT-PCR kits were obtained from Bio-Rad and Applied Biosystems, Waltham, MA, USA.

### 4.2. Animals

Wistar male rats (200–250 g) were purchased from the MRC (Medical Research Center), Jazan University. The animals were retained in a polycarbonate (box) cage with bedding made of fine rice husk. Before the experiment, the animals were housed in the College of Pharmacy’s animal house for a week in order to acclimate to the conditions, which met international standards for laboratory environments and a 25 ± 2 °C temperature, 45–55% humidity, unlimited access to food, water, and 12 h cycles of light and darkness. The study was approved by the Institutional Research Review and Ethical Committee (IRREC), Jazan University (approval No.: 724/216/1443). On the day of sacrifice, the food was removed overnight, an estimated 12 h prior to sacrificing.

### 4.3. Nano Formulation of Piperine (NP)

A commonly used hot homogenization method prepared nanoformulations containing piperine (NP). Liquid lipid (sesame oil), solid lipid (stearic acid) and surfactants such as sodium dodecyl sulfate and tween-80 were used in the preparation of lipid nanoparticles, as reported in our earlier study [24].

### 4.4. Dose Selection

Cyp and saline were freshly prepared daily and given at a dose of 50 mg/kg/day [44,45] orally for 15 days. NP was given for 10 days orally with varying doses (250, and 500 μg/kg/day) and Cyp treatment commenced 5 days later. We maintained a one-hour time gap between NP and Cyp doses.

### 4.5. Experimental Design

Rats were divided into six groups, with six in each group, and the division was random. Group I was treated as a control and received only saline, while groups III–V were treated with Cyp (50 mg/kg) daily for 15 days. Following five days of Cyp treatment, groups (IV–V) next received daily doses of NP 250 and 500 μg/kg/day separately. Group II, which was treated with only NP for 10 days, received NP at a dose of 500 μg/kg. Toxicity indicators were seen throughout therapy. The retro-orbital venous plexus was applied to pooled blood at the end of the experiment to separate the serum for biochemical evaluation. Following this, we sacrificed all rat group, and rat livers were removed and tested for oxidative stress markers. Tissues were kept in 10% formalin for histopathological examinations. 

### 4.6. Sample Preparation

Ocular punctures were used to collect blood samples the from respective groups. Serum was extracted from blood samples via the centrifugation method. The animals were then given an intramuscular injection of ketamine hydrochloride (30 mg/kg b/w) to induce anesthesia before being terminated via cervical dislocation. Quickly removing the liver tissue allowed for the preparation of homogenates and post-mitochondrial supernatants (PMSs). An ultraviolet (UV) spectrophotometer (UV-1800, Shimadzu, Kyoto, Japan) was used to perform the biochemical experiment in homogenate/PMS.

### 4.7. Serum Biochemical Assay

Liver function-specific serum markers (alanine aminotransferase (ALT), aspartate aminotransferase (AST), alkaline phosphatase (ALP), total protein, and albumin) were assessed using a spectrophotometer as per the manufacturer’s guidelines.

### 4.8. Estimation of LPO and GSH

Lipid peroxidation was assayed in tissue homogenate using the methodology of Hussain et al. [46]. GSH estimation by Ashfaq et al. [29] was performed using SSA (4%) in a 1:1 ratio with a sample to assess the GSH. The solution combination was centrifuged for 15 min at 3000 rpm after being stored at 4 °C for an hour in order to separate the supernatant. The absorbance was recorded at 535 nm and 412 nm, respectively.

### 4.9. Estimation of Activity of SOD

By tracking the auto-oxidation of epinephrine [47] at pH 10.4 for three minutes at 480 nm, SOD was assessed. The sample contained (50 mM, pH 10.4) a glycine buffer and 0.2 mL of liver tissue PMS. At 480 nm, the optical density was measured. SOD was given as nmol of protected epinephrine per minute per milligram of protein.

### 4.10. Estimation of Catalase Activity

According to Alshahrani et al. [48], 3000 µL of the reaction mixture, containing 1000 µL of H_2_O_2_, 50 μL PMS, and 1950 µL of phosphate buffer, was used to measure catalase activity. At 240 nm, the assay sample’s reading was calculated using a kinetic technique. Catalase activity is measured in nmol of H_2_O_2_ per minute per milligram of protein. 

### 4.11. Estimation of Interleukin Cytokines (IL-6 and IL-1β) and Apoptotic Markers (Caspase-3 and -9)

These steps were carried out according to the manufacturer’s protocols. As directed by the manufacturer (Abcam, Cambridge, UK), an ELISA reader (BioTek, Winooski, VT, USA) was used to measure optical density at 450 nm and 405 nm, respectively. Picograms of antigen per milligram of protein were used to express tissue cytokine and caspase concentrations.

### 4.12. DNA Damage and Apoptosis Assay

DNA-QIAGEN kits were used to extract liver DNA (25 mg tissue). Spectrophotometers were used to further measure the separated DNA samples, as previously mentioned [42]. Samples of 10 μL were taken from each group and placed onto a gel prepared by 2% agarose with 0.5 g/mL of ethidium bromide in order to analyze DNA fragmentation. The DNA sample was electrophoresed and analyzed using a UV trans-illuminator. 

### 4.13. Quantitative Real Time Polymerase Chain Reaction

An RNA Mini Kit (Cat#7326820, BIO-RAD) was used to isolate RNA from the tissue. cDNA was synthesized from RNA using a High-Capacity cDNA Reverse Transcription Kit with an RNase Inhibitor (Applied Biosystems A25918). The absorption at 260 nm was measured in order to quantify RNA and cDNA. The generated cDNAs and SYBR green dye with primers of NF-k, Bax, IL-1β, IL-6, and caspase-3 and -9 (Table 4) were used in qRT-PCR for gene expression by Real-Time PCR System (BIO-RAD). Gene expressions were given as relative expressions using the 2∆∆CT method, using B-actin as standard [49].

### 4.14. Liver Histopathological Studies

The liver tissue was washed with physiological saline (0.89%), and then fixed for 48 h in a solution of formaldehyde (40%) glacial acetic acid, and then treated with methanol (1:1:8, *v*/*v*). Beeswax paraffin was used to embed fixed tissue and tissue blocks were created. The samples were taken from paraffin-embedded tissues and were 5 nm thick. Additional portions underwent deparaffinization with xylene, ethanol, and water exchange. Hematoxylin and eosin were used to stain the deparaffinized slices, which were mounted on cover slides. A light microscope was used to take pictures. The entire procedure is a small modification of the Ashafaq et al. [50] approach.

### 4.15. Estimation of Protein

Protein values from all group were assessed as stated by Lowry et al. [51]. The result of protein assessment were calculated using BSA as the reference.

### 4.16. Statistical Analysis

The results were contrasted among each other and stated as mean ± SEM. Analysis of variance (ANOVA) and Tukey’s Kramer were used to statistically analyze the data. Data were deemed statistically significant when *p* > 0.05.

## Figures and Tables

**Figure 1 ijms-24-15361-f001:**
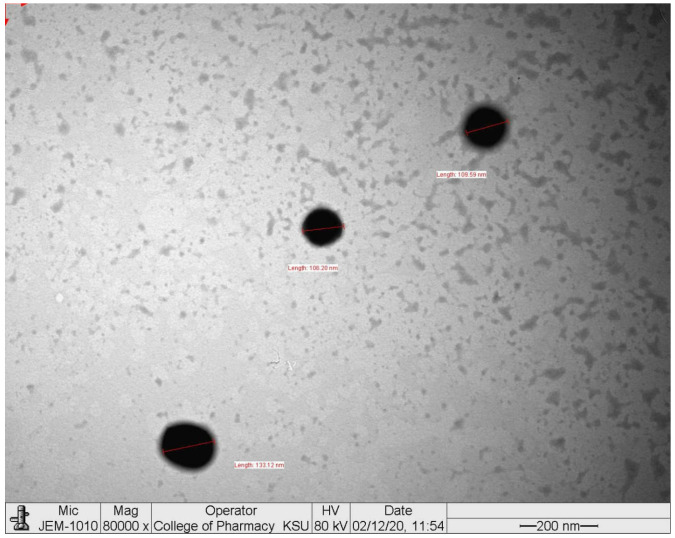
Image of NP acquired by transmission electron microscopy.

**Figure 2 ijms-24-15361-f002:**
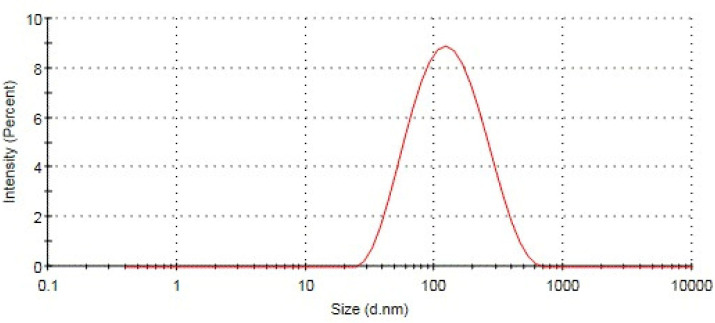
Particle size distribution of NP. Average particle size distribution (*n* = 1) of optimized formulations of NP. The particle size was 151 nm with a narrow size distribution (<0.5).

**Figure 3 ijms-24-15361-f003:**
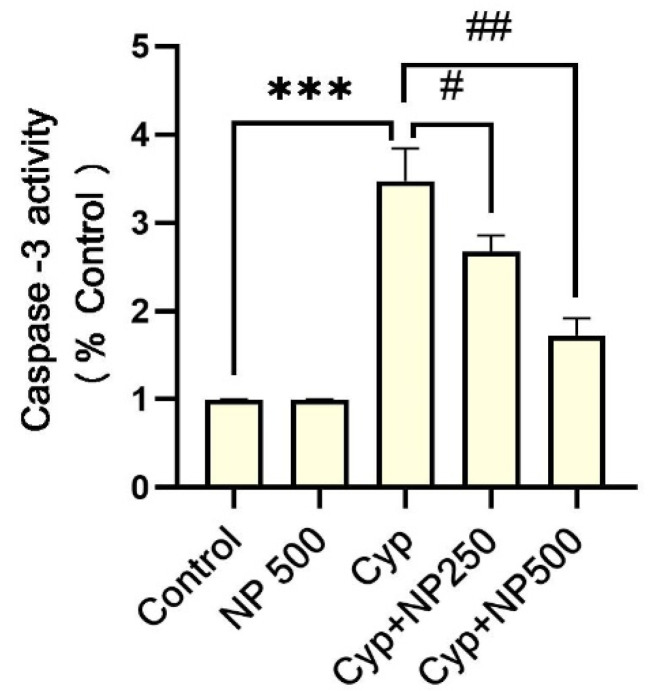
Effect of NP treatment on caspase-3. Value shown as mean ± SEM (*n* = 5). *** *p* < 0.001 vs. control, **^#^**
*p* < 0.05, **^##^***p* < 0.01 vs. Cyp group.

**Figure 4 ijms-24-15361-f004:**
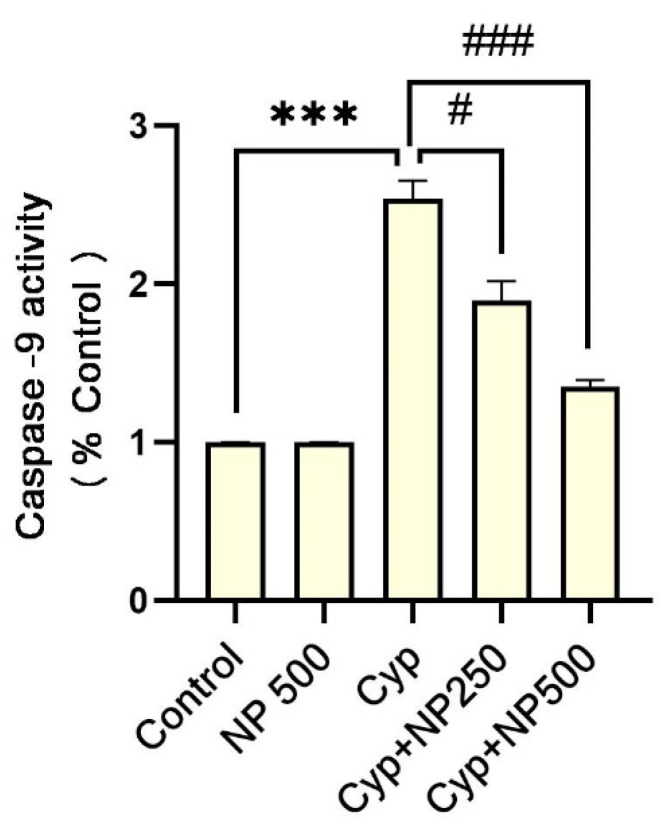
Effect of NP treatment on caspase-9. Value shown as mean ± SEM (*n* = 5). *** *p* < 0.001 vs. control, **^#^**
*p* < 0.05, **^###^**
*p* < 0.001 vs. Cyp group.

**Figure 5 ijms-24-15361-f005:**
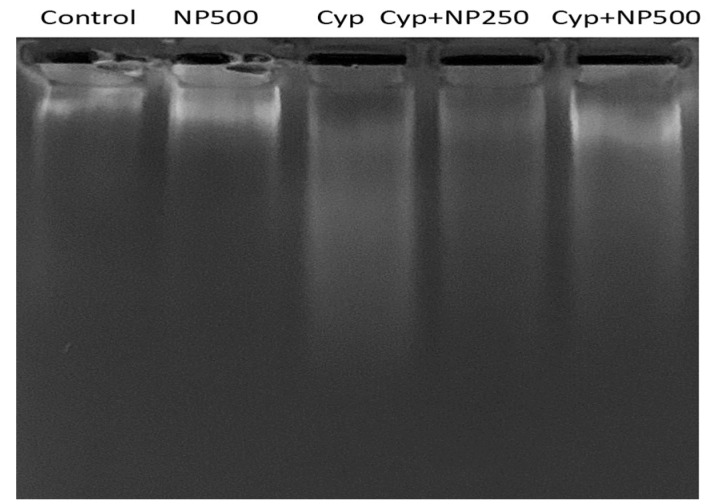
NP administration successfully protected the liver DNA fragmentation. Lane-1: control; lane-2: NP500; lane-3: Cyp; lane-4 Cyp + NP250 and lane-5: Cyp + NP500.

**Figure 6 ijms-24-15361-f006:**
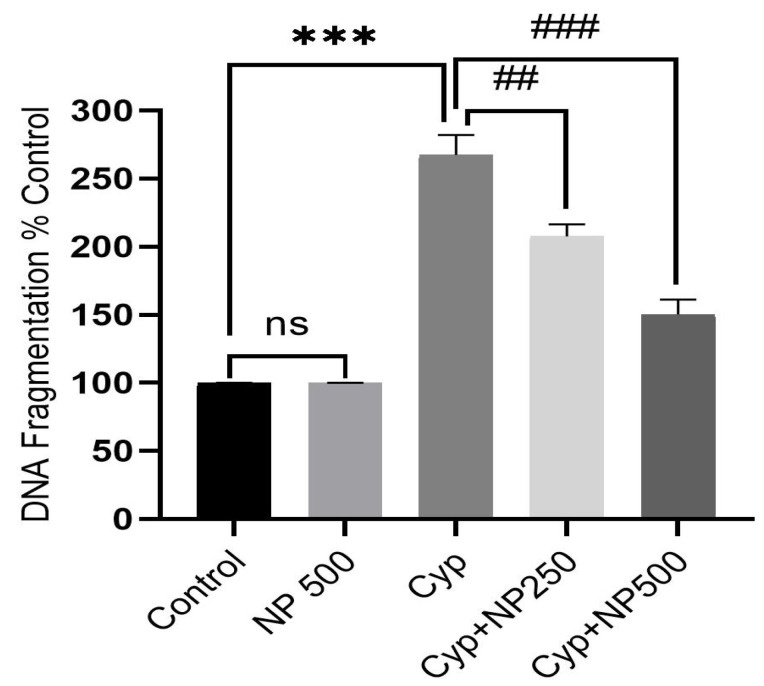
NP treatment positively reduces apoptosis against Cyp Prompted DNA Fragmentation. Value shown as mean ± SEM with *n* = 5 animals in a group. *** *p* < 0.001 vs. control; **^##^**
*p* < 0.01, **^###^**
*p* < 0.001 vs. Cyp; ^ns^ *p* vs. control (ns = non-significant).

**Figure 7 ijms-24-15361-f007:**
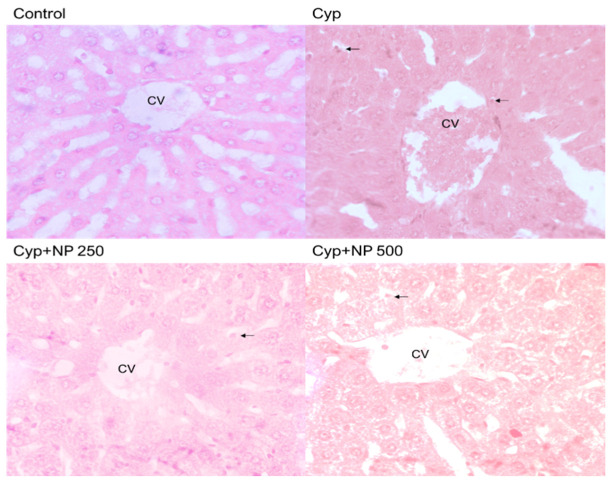
Liver histology of control, Cyp-treated and Cyp + NP group. Sections of the liver tissue: denote control with healthy cell morphology and central hepatic vein. Represents the Cyp treatment, showing congestions in the central vein (cv) and gathering of inflammatory cells in the portal vein along with pyknotic nuclei (black arrow). Represents Cyp + NP 250- and 500-treated groups respectively. These show mild-to-moderate cellular damage with improved cell architecture, which induces hepatoprotective effects of NP against Cyp toxicity. Magnification 40×.

**Figure 8 ijms-24-15361-f008:**
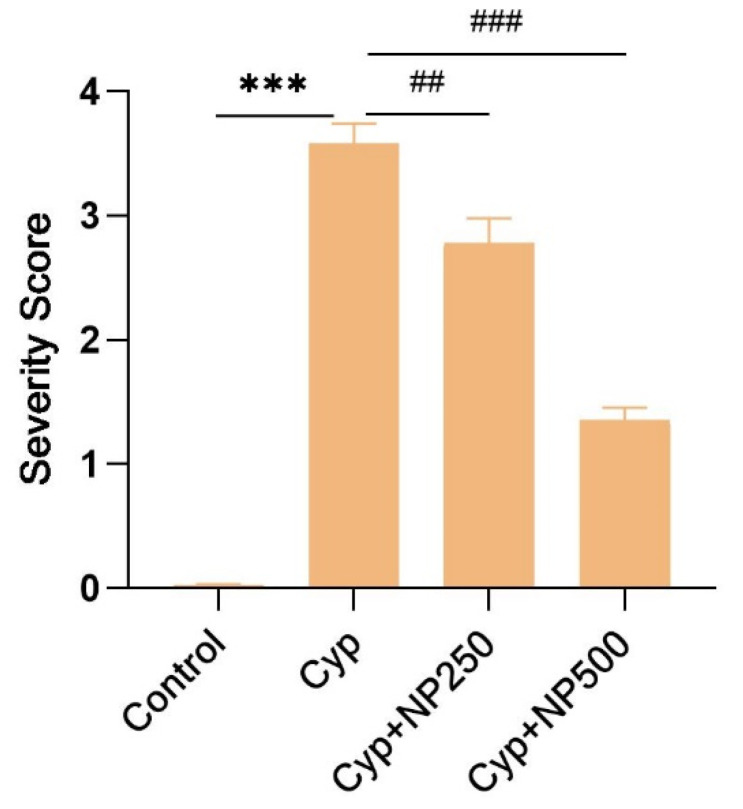
Graph showing severity score (0–4) of liver histology. *** *p* < 0.001 vs. control; **^##^**
*p* < 0.01, **^###^**
*p* < 0.001 vs. Cyp.

**Figure 9 ijms-24-15361-f009:**
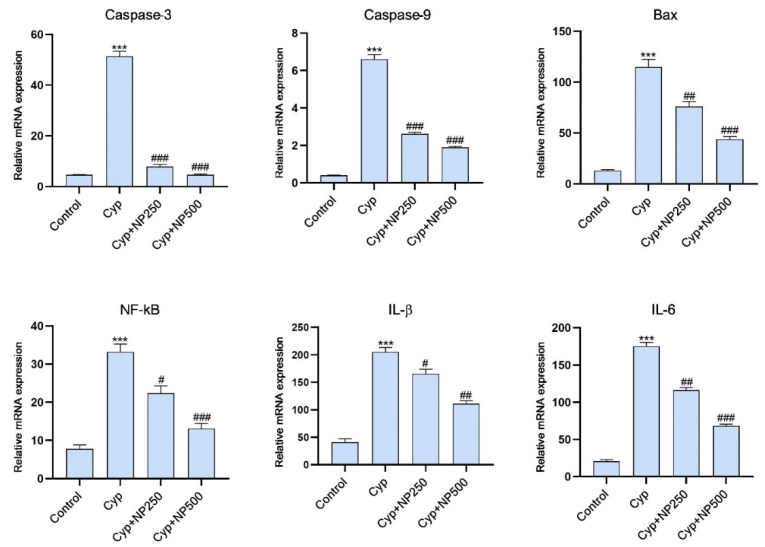
Outcome of NP versus Cyp treatment on mRNA expression level of caspase 3 & 9, Bax, NF-kB, IL-1β and IL-6. Data were expressed as the mean ± SE (*n* = 3). *** *p* < 0.001 vs. control, **^#^**
*p* < 0.05; **^##^**
*p* < 0.01; **^###^**
*p* < 0.001 vs. Cyp.

**Table 1 ijms-24-15361-t001:** Effect of NP treatment on Cyp-induced changes in Serum Markers in rat liver.

Groups	ALT	AST	ALP	Protein	Albumin
Control	40.52 ± 5.63	37.11 ± 2.92	92.883 ± 9.675	8.882 ± 0.067	4.36 ± 1.15
NP500	44.378 ± 6.52	31.073 ± 0.983	85.614 ± 3.018	7.31 ± 0.1.28	4.07 ± 0.88
Cyp	179.46 ± 12.29 ***	81.051 ± 7.35 ***	181.831 ± 1.527 ***	4.01 ± 0.052 **	2.19 ± 1.86 **
Cyp + NP 250	158.51 ± 9.84 ^##^	51.66 ± 4.42 ^##^	149.62 ±11.04 ^###^	6.37 ± 0.48 ^#^	3.01 ± 1.05 ^##^
Cyp + NP 500	95.66 ± 8.42 ^###^	44.56 ± 5.37 ^###^	131.79 ±11.53 ^###^	7.85 ± 1.03 ^##^	3.25 ± 2.13 ^###^

Value expressed as mean ± SE (*n* = 6). ** *p* < 0.01; *** *p* < 0.001 vs. control group. ^#^
*p* < 0.05; ^##^
*p* < 0.01; ^###^
*p* < 0.001 vs. Cyp-treated group.

**Table 2 ijms-24-15361-t002:** Effect of NP treatment on Cyp-induced oxidative stress rat liver.

Groups	MDA (nmol/g Tissue)	GSH (DTNB Conjugate Formed/mg Protein)	SOD (nmol Epinephrine Protected from Oxidation/min/mg Protein)	CAT (nmol of H_2_O_2_ Consumed/min/mg Protein)
Control	1.148 ± 0.059	28.117 ± 1.056	25.883 ± 2.675	5.882 ± 0.067
NP500	1.198 ± 0.068	29.073 ± 0.983	26.614 ± 3.018	6.001 ± 0.083
Cyp	2.081 ± 0.161 ***	17.051 ± 1.681 ***	14.831 ± 1.527 ***	3.181 ± 0.052 **
Cyp + NP250	1.781 ± 0.089 ^#^	21.129 ± 1.671 ^#^	19.390 ± 2.118 ^#^	4.173 ± 0.064 ^#^
Cyp + NP500	1.354 ± 0.087 ^###^	24.280 ± 1.143 ^##^	23.137 ± 2.482 ^##^	4.951 ± 0.48 ^##^

Value expressed as mean ± SE (*n* = 6). ** *p* < 0.01; *** *p* < 0.001 vs. control group. ^#^
*p* < 0.05; ^##^
*p* < 0.01; ^###^
*p* < 0.001 vs. Cyp-treated group.

**Table 3 ijms-24-15361-t003:** Effect of NP treatment on Cyp-induced interleukins in rat liver.

Groups	IL-1β (pg/mL)	IL-6 (pg/mL)	TNF-α (pg/mL)
Control	224.67 ± 20.56	281.40 ± 21.63	236.75 ± 17.96
NP500	223.98 ± 21.35	282.67 ± 19.43	235.88 ± 23.48
Cyp	845.19 ± 32.72 ***	910.27 ± 36.79 ***	518.07 ± 33.79 ***
Cyp + NP250	571.34 ± 26.37 ^##^	710.91 ± 31.58 ^#^	362.43 ± 28.56 ^##^
Cyp + NP500	411.95 ± 18.26 ^###^	522.64 ± 27.11 ^###^	259.21 ± 19.74 ^###^

Value expressed as mean ± SE (*n* = 6). *** *p* < 0.001 vs. control group. ^#^
*p* < 0.05; ^##^
*p* < 0.01; ^###^
*p* < 0.001 vs. Cyp-treated group.

**Table 4 ijms-24-15361-t004:** List of primers sequences.

Oligo	Primer Sequences	Tm (c)
IL-1beta	5′-ATGGCAACCGTACCTGAACCCA-3′ F 5′-GCTCGAAATGTCCCAGGAA-3′ R	64.0 58.4
IL-6	5′-AGTTGCTTCTTGGGACTGATGT-3′ F 5′-TGCTCTGAATGACTCTGGCTTTG-3′ R	62.9 62.9
NF-kB	5′-CATGAAGAGAAGACACTGACC-3′ F 5′-TGGATAGAGGCTAAGTGTAGA-3′ R	59.4 57.4
Bax	5′-GCCTCCTTTCCTACTTCGGG-3′ F 5′-CTTTCCCCGTTCCCCATTCA-3′ R	62.5 60.5
Caspase-9	5′-TGTCCTACTCTACTTTCCCAGGT-3′ F 5′- GTGAGCCCACTGCTCAAAGAT -3′ R	62.5 60.5
Caspase-3	5′-GCTACGATCCACCAGCATTT-3′ F 5′-ATGCCACCTCTCCTTTCCTT-3′ R	58.4 58.4
Beta-actin	5′-CAACCTTCTTGCAGCTCCTC-3′ F 5′-TTCTGACCCATACCCACCAT-3′ R	60.5 58.4

## Data Availability

The data available within the article.

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
