# Peer review of "Hepatoprotective and Antioxidant Effects of Nanopiperine against Cypermethrin via Mitigation of Oxidative Stress, Inflammations and Gene Expression Using qRT-PCR"

_ijms, 2023, doi:10.3390/ijms242015361_

Round 1
Reviewer 1 Report
The entitled manuscript" Hepatoprotective Effect of Nanopiperine Against Cypermethrin Toxicity Through Inflammatory Cytokines and Gene Expression by qRT-PCR. " by Sohail Hussain et al. was aimed to study to evaluate the hepatoprotective activities of nano-piperine (NP) against Cyp toxicity by qRT-PCR . In addition to the qRT-PCR method, the authors also use transmission electron microscopy, Serum Biochemical Assay, SOD and Catalase activity assays, and Histopathological tests.
I consider this manuscript to be original and relevant, contributing to the field's results and hepatology research. In my opinion, the results strengthen and support the conclusion of the article.
I also found the number of references adequate and convincing.
The article includes 9 figures illustrating the results and 4 tables.
I accept the article in its current form.
Author Response
The authors would like to thank reviewers for critically reviewing our manuscript.
Reviewer 2 Report
The article proposed by the authors for publication, I think is very important, especially in the current context where there is a risk that different types of food are contaminated with chemical substances generated by the artificial feeding of the soil or the use of pesticides in order to destroy different types of pests. The aim of the work is to evaluate the hepatoprotective capacity of a nano-piperine against cypermethrin, a synthetic pyrethroid frequently used in agriculture.
Research evaluation:
- The introduction provides information about liver pathologies generated by chemical contaminants from different food sources; pharmacotoxicological data on the pyrethroid cypermethrin; pharmacotherapeutic data on piperine, active principle of Piper nigrum; data on the importance of nanoformulations in increasing the bioavailability of natural phytocompounds;
- The results are presented in detail; for their easy understanding and interpretation, the authors summarize the results in tables, graphs, diagrams, histograms, photomicrographs; all results are processed statistically;
- In the Discussions section, the authors correlate all the results obtained with the data from the specialized literature related to the research carried out; it is a justifying interdependence of all the data obtained with those presented in the specialized literature;
- The section in which the authors present materials and methods of analysis I believe is rendered properly;
- The bibliography is justifiable.
I emphasize the complexity of the study.
Author Response
Response to Reviewers Comments:
The authors would like to thank reviewers for critically reviewing our manuscript. We have tried to incorporate all the changes suggested by the reviewers. The responses to the comments are as follows:
Reviewer 2
Comments and Suggestions for Authors
The article proposed by the authors for publication, I think is very important, especially in the current context where there is a risk that different types of food are contaminated with chemical substances generated by the artificial feeding of the soil or the use of pesticides in order to destroy different types of pests. The aim of the work is to evaluate the hepatoprotective capacity of a nano-piperine against cypermethrin, a synthetic pyrethroid frequently used in agriculture.
Research evaluation:
- The introduction provides information about liver pathologies generated by chemical contaminants from different food sources; pharmacotoxicological data on the pyrethroid cypermethrin; pharmacotherapeutic data on piperine, active principle of Piper nigrum; data on the importance of nanoformulations in increasing the bioavailability of natural phytocompounds;
- The results are presented in detail; for their easy understanding and interpretation, the authors summarize the results in tables, graphs, diagrams, histograms, photomicrographs; all results are processed statistically;
- In the Discussions section, the authors correlate all the results obtained with the data from the specialized literature related to the research carried out; it is a justifying interdependence of all the data obtained with those presented in the specialized literature;
- The section in which the authors present materials and methods of analysis I believe is rendered properly;
- The bibliography is justifiable.
I emphasize the complexity of the study.
Response: The authors would like to express their gratitude and appreciation to the reviewer who took the time to critically review our work.

Reviewer 3 Report
Please find the comments in the attached file.

Author Response
The authors would like to thank reviewers for critically reviewing our manuscript. We have tried to incorporate all the changes suggested by the reviewers. The responses to the comments are as follows:
